# Offshore Occurrence of a Migratory Bat, *Pipistrellus nathusii*, Depends on Seasonality and Weather Conditions

**DOI:** 10.3390/ani11123442

**Published:** 2021-12-02

**Authors:** Sander Lagerveld, Bob Jonge Poerink, Steve C. V. Geelhoed

**Affiliations:** 1Wageningen Marine Research, Ankerpark 27, 1781 AG Den Helder, The Netherlands; steve.geelhoed@wur.nl; 2Ecosensys, Hoofdweg 46, 9966 VC Zuurdijk, The Netherlands; bob.jongepoerink@ecosensys.nl

**Keywords:** migration, offshore wind farms, Nathusius’ pipistrelle, acoustic monitoring

## Abstract

**Simple Summary:**

Migratory bats regularly fly over the North Sea, where the number of offshore wind farms will increase rapidly in the next decades. Information is urgently needed on the timing and the conditions bats can be expected offshore, since windfarms can cause fatalities amongst bats. We therefore collected acoustic data on the presence of bats at four nearshore locations at sea between 2012 and 2016. Modelling the occurrence of Nathusius’ pipistrelle for 480 nights in autumn showed that its migration is strongest in early September, with east-northeasterly tailwinds, low wind speeds, and relatively high temperatures. The species’ migration did not show a strong relationship with other factors, i.e., moon phase, cloud cover, atmospheric pressure, rain, and visibility. Our results provide valuable input to policy-makers to prescribe mitigation measures to reduce bat fatalities in offshore wind farms.

**Abstract:**

Bats regularly migrate over the North Sea, but information on the environmental conditions when this occurs is scarce. Detailed information is urgently needed on the conditions under which bats can be expected offshore, as the number of offshore windfarms that can cause fatalities amongst bats in the North Sea is increasing rapidly. We performed ultrasonic acoustic monitoring at multiple nearshore locations at sea between 2012 and 2016 for, in total, 480 monitoring nights. We modelled the offshore occurrence of Nathusius’ pipistrelle in autumn as a function of weather conditions, seasonality, and the lunar cycle using a generalized additive mixed model (GAMM). We investigated which covariates are important using backward selection based on a likelihood ratio test. Our model showed that important explanatory variables for the offshore occurrence of Nathusius’ pipistrelle are seasonality (night in year), wind speed, wind direction, and temperature. The species’ migration is strongest in early September, with east-northeasterly tailwinds, wind speeds < 5 m/s, and temperatures > 15 °C. Lunar cycle, cloud cover, atmospheric pressure, atmospheric pressure change, rain, and visibility were excluded during the model selection. These results provide valuable input to reduce bat fatalities in offshore wind farms by taking mitigation measures.

## 1. Introduction

The development of offshore wind energy production in the North Sea has grown extensively since the early 2000s. In Dutch waters, a capacity of 2.5 GW has been realized until 2021 and this is planned to grow to 11.5 GW in 2030. By then, approximately 1100 offshore wind turbines will produce 49 TWh annually, which corresponds to 40% of the current Dutch national electricity consumption. Offshore wind energy production is therefore expected to play a significant role in the reduction of greenhouse-gas emissions. At the same time, however, wind energy production causes biodiversity loss due to mortality, habitat loss, and barrier effects [1,2]. One of the major concerns is mortality amongst bats due to collisions and possibly barotrauma [3,4,5,6,7,8,9,10,11], which affects both migratory and local populations [12,13]. For wind farms on land, it is estimated that 250,000 bats are likely killed annually in Germany [14], whilst 600,000 bat fatalities have been reported in the USA in one year [15].

There are several options to reduce the number of bat fatalities [16]. An important factor which determines the fatality risk is the location choice of wind farms. High-risk areas are coasts and forested hills, whereas mortality is relatively low in agricultural areas further away from the coast [7]. Effective mitigation has been achieved by curtailment procedures that limit the production time of wind turbines [16,17,18,19,20]. Increasing the cut-in speed at which blades start rotating to wind speeds of 5 m/s has resulted in an average reduction of 62% in the number of fatalities [20]. Finally, the number of fatalities may be reduced by the application of deterrent devices [21,22].

Research on the occurrence of bats has shown that bats regularly may occur offshore in the North Sea area [23,24,25,26,27,28]. As bats show similar foraging behavior around offshore wind turbines as around onshore wind turbines [29], it seems likely that fatalities also occur at sea.

The most frequently encountered species over the North Sea is Nathusius’ pipistrelle *Pipistrellus nathusii*, but common pipistrelle *P. pipistrellus*, common noctule *Nyctalus noctula*, Leisler’s bat *N. leisleri*, particolored bat *Vespertilio murinus*, Northern bat *Eptesicus nilssonii*, and Serotine bat *E. serotinus* are also recorded [23,24,25,26,27,28]. Most bats are recorded during the migration seasons, from late March until June and from late August until October. Yet, it is unknown whether migrant bats cross the North Sea in a broad front, like many songbirds do [30,31], or if they show spatially distinct migration patterns. If so, the suitability of areas for offshore wind developments may be identified based on relationships between bat activity, distance from shore, and weather patterns. Currently, curtailment is the only mitigation measure for bats that has been implemented in the Dutch North Sea.

Relatively few studies have been published on the conditions under which bats can be expected offshore over the last years [23,24,25,26,27,28]. These studies, however, are based on small datasets, and some seem to contradict each other. Furthermore, studies on the spatial and temporal offshore occurrence of bats are virtually lacking. Policy-makers urgently need detailed information on when and under which conditions bats can be expected in offshore wind farms. This will enable them to take the occurrence of bats into account during marine spatial planning and when developing mitigation measures for offshore wind farms.

We performed ultrasonic acoustic monitoring in three different wind farms in the Dutch North Sea during five consecutive years. We analyzed the occurrence of Nathusius’ pipistrelle in autumn in relation to weather parameters (wind speed, wind direction, visibility, atmospheric pressure, atmospheric pressure change, rain, cloud cover, and temperature), seasonality (night in year), and moon illumination. This study shows that Nathusius’ pipistrelle chooses specific environmental conditions for departures over sea during autumn migration.

## 2. Materials and Methods

### 2.1. Study Area

The study area is located off the western North Sea coast of the Netherlands. The area includes Offshore Wind Farm Egmond aan Zee (OWEZ), Princess Amalia Wind Farm (PAWP), and Luchterduinen Wind Farm (LUD). Figure 1 shows the study area with the locations where acoustic bat monitoring was executed. At Offshore Wind Farm Egmond aan Zee, monitoring was performed at the meteorological mast, at Princess Amalia Wind Farm, at wind turbine 22 and at the offshore high-voltage station (OHVS), and at Luchterduinen Wind Farm, at the OHVS.

**Figure 1 animals-11-03442-f001:**
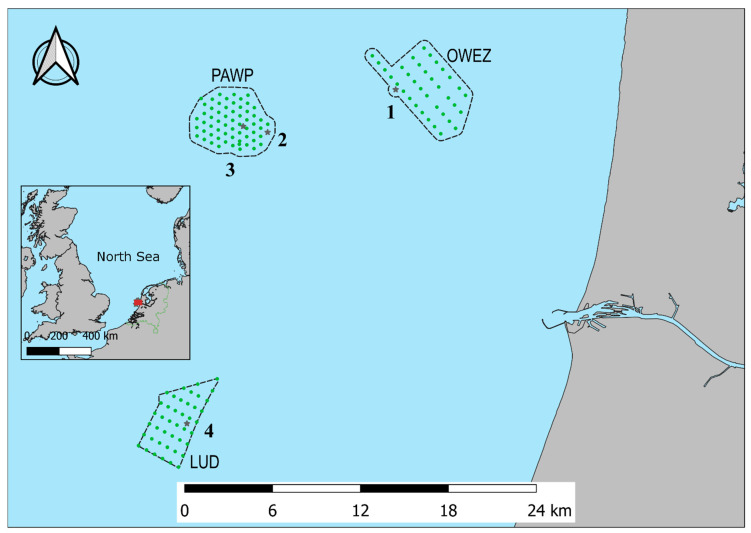
Location of the acoustic bat monitoring stations in Offshore Wind Farm Egmond aan Zee (OWEZ), Princess Amalia Wind Farm (PAWP), and Luchterduinen Wind Farm (LUD) along the western coast of the Netherlands. Wind turbines are indicated by green circles. See Table 1 for details on the monitoring locations, indicated by black stars.

### 2.2. Equipment

We monitored the acoustic bat activity with one ultrasonic recorder (Batcorder-EcoObs GmbH) per monitoring location. Stored in a waterproof box, the recorders were attached to the locations at a height of 15 m above sea level. The recorders were set to record only after being triggered by a bat call or by a bat call-like sound. All sounds in the range of 16–150 kHz were recorded, with the date and time of the recording. Sounds are typically recorded at a distance of 15–100 m from the recorder depending on species-specific echolocation characteristics, the environmental conditions, and the recorder settings [32]. We set the threshold amplitude of the recorder to −36 dB in order to boost microphone sensitivity. At this sensitivity level, the maximum detection distance of Nathusius’ pipistrelle is approximately 30 m. Default settings of the manufacturer were used for the other parameters, i.e., post-trigger: 400 m, threshold frequency: 16 kHz, recording quality: 20, and noise filter: 1.

Daily status updates allowed monitoring the performance of the recorders throughout the season. The microphones were replaced in March and July each year or sooner when microphone sensitivity levels were continuously low. All microphones were recalibrated after use to ensure the consistency of the measurements. Monitoring took place between August 2012 and October 2016 (Table 2).

### 2.3. Data Preparation

Echolocating bats emit ultrasonic pulses to gain information about their environment. Maintenance or production activities at offshore structures, however, also produce ultrasonic sounds. We used BcAdmin 2.0 (EcoObs GmbH, Nürnberg, Germany) to separate the sound files containing bat calls from those with ‘noise’. Each bat call recording was analyzed and identified to the lowest taxonomic level as possible using the criteria provided by [32].

Weather data from the KNMI offshore weather stations 203 PB-11B (52°35′ N, 3°33′ E) and 212 Hoorn-A (52°91′ N, 4°15′ E) were obtained from the Royal Netherlands Meteorological Institute (www.knmi.nl), retrieved on 29 May 2017. Weather variables included in the dataset are: wind direction and wind speed averaged over 10 min measured at an altitude of 10 m above sea level, temperature and relative humidity at 1.5 m height, atmospheric pressure at sea level, horizontal visibility in meters, cloud cover in octants, ranging from 0 (no clouds) to 8 (completely overcast), and rain. For the latter variable, a value of 1 indicates rain did occur in the preceding hour, while 0 indicates it did not. Horizontal visibility was calculated as the numerical midpoint of the given range.

Because weather data from a single weather station showed gaps, we averaged the data of the two stations after removing the missing data. If data from both stations were missing, the value was qualified as missing. The average of each weather parameter is the arithmetic mean over the period between sunset and sunrise, except for wind direction. The average of this parameter was calculated by:a_mean = atan2(sum_i(sin(a_i)), sum_i(cos(a_i))) mod 2 × pi(1)
where a_mean is the wind direction measured in radians, a_i is the i_th wind direction to be averaged (measured in radians), and atan2 is the arctangens function with two arguments, as implemented in the R programming language.

The lunar phase was calculated using the R-package lunar [33].

### 2.4. Statistical Analysis

We exclusively used acoustic data from Nathusius’ pipistrelle for the statistical analysis. Since bats are nocturnal, their occurrence was analyzed per night. We used the presence per night as the response variable for the analysis from mid-August (night number 232) until mid-October (night number 286). The presence per night (Y_i) is either 1 when one or more bats have been detected or 0 when bats have not been detected. We assumed that the presence per night is Bernoulli-distributed with probability Pi_i.
Y_i ~ Bernoulli(Pi_i)(2)

The mean and variance of a Bernoulli-distributed variable are, respectively:E(Y_i) = Pi_i(3)
var(Y_i) = Pi_i ∗ (1 − Pi_i)(4)

In order to investigate spatiotemporal patterns, we modelled the response variable as a function of the covariates, using a logistic link function to ensure that the fitted probabilities were between 0 and 1.
logit(Pi_i) = Intercept + Covariates(5)

Covariates included in the analysis were XY coordinates [UTM], night in year, lunar cycle [%], cloud cover [octant], atmospheric pressure [mB], atmospheric pressure change [mB], fraction of hour intervals with rain [%], temperature [°C], visibility [km], humidity [%], wind direction [degrees], and wind speed [m/s]. All fixed covariates were continuous. 

We used the protocol provided by [34] as guidance for the analysis in R [35]. Cleveland dotplots were used to assess outliers in the covariates. Collinearity between the continuous covariates was assessed with multi-panel scatterplots, Pearson correlation coefficients and variance inflation factors. The relationships between the response variable and the continuous covariates were checked with multi-panel scatterplots.

We used a generalized additive mixed model (GAMM) for the analysis using the R-package GAMM 4 [36] and used the monitoring location as a random effect. To assess potential spatial patterns, we included a tensor product smoother for the XY coordinates. To capture seasonal patterns, the covariate night in year was included with a (default) thin-plate regression spline smoother in the model, and the covariates lunar cycle and wind direction were incorporated as cyclic smoothers (see [37] for background information). The other continuous covariates were included as linear covariates. All fixed covariates were standardized to avoid numerical problems.

In order to reduce the model complexity, we investigated which covariates in the fixed structure were important, using a stepwise backward selection based on a likelihood ratio test [38]. Once the optimal model was found, a model validation was applied where we plotted the Pearson residuals against fitted values, and against each covariate both in the model and not in the model. In addition, potential spatial and temporal autocorrelation in the Pearson residuals was assessed with variograms.

## 3. Results

### 3.1. Monitoring Data

From mid-August to mid-October, 480 nights were monitored in 2012–2016 (Table 2). The percentage of nights with recorded bat activity of Nathusius’ pipistrelle amounted to 25% at OWEZ, 21% at PAWP WT22, 15% at PAWP OHVS, and 11% at LUD OHVS.

Figure 2 shows an example of the recorded bat activity of Nathusius’ pipistrelle per 1 min intervals throughout the night at Offshore Wind Farm Egmond aan Zee in autumn 2013. Some nights contained only one 1 min interval with bat activity, whereas in other nights, multiple 1 min intervals occurred. When multiple time intervals occur sequentially, it is likely that the same individual is involved. Note, however, that multiple individuals can be present at the same time, as up to three individuals were recorded simultaneously at Offshore Wind Farm Egmond aan Zee in 2012 [26].

The dataset comprised 480 monitoring nights, of which two records with missing weather data were removed. There were no obvious outliers in the covariates. The covariate *humidity* was removed from the dataset as it was collinear with the covariate *visibility*. After this removal, all variance inflation factors were well under 3. The response variable was zero-inflated (82%), but, as a Bernoulli distribution was chosen, this amount of zeros was no immediate concern for the analysis.

A stepwise backward selection based on a likelihood ratio test [38] resulted in consecutively excluding the covariates XY coordinates, cloud cover, lunar cycle, atmospheric pressure, fraction of hours per night with rain, visibility, and atmospheric pressure change from the model. The optimal model therefore resulted to be:logit(Pi_i) = Intercept + wind speed + s(wind direction) + s(night in year) + temperature(6)

The properties of parametric coefficients and smooth terms are shown in Table 3.

The model validation did not indicate violations of the model assumptions on independence, heterogeneity, and normality.

### 3.2. Graphical Representation of the Model

To visualize the influence of individual covariates, we calculated the predicted values from the model using a range of values between the minimum and the maximum observed values of that particular covariate and the mean value of the other covariates. Using ggplot2 [39], we plotted the actual monitoring data as well as the predicted values including their 95% confidence intervals (Figure 3a–d). The probability of presence of bats decreased with a higher *wind speed* (Figure 3a). It should be noted that wind speeds <3 m/s (5% of the nights) and >12m/s (5% of the nights) rarely occurred in the study period. During nights when bats were present, wind speeds < 5 m/s were recorded in 67% of our observations, an additional 31% of observations occurred with wind speeds of 5–8 m/s, and the remaining 2% with wind speeds > 8 m/s.

Wind direction had a marked influence on the probability of bat presence, with a distinct peak at 64° (ENE), and the lowest values between 180° and 360° (S via W to N, Figure 3b). Figure 3c shows the seasonal pattern. The probability of bat presence increased from the start of the modelled period around mid-August (night in year = 230) to a single peak in early September (night in year = 249) and gradually petered out in mid-October (night in year = 285). The probability of bat presence increased with higher temperatures (Figure 3d). During nights with an average temperature < 15 °C, 11% of the observations were reported, whilst nights with an average temperature between 15 and 18 °C and >18 °C accounted for 50 and 39% of our observations, respectively.

## 4. Discussion

Our study yields valuable information on the offshore occurrence of Nathusius’ pipistrelle that can be used as input for mitigation measures preventing potential collisions of bats with offshore wind turbines.

We found a temporal pattern in occurrence, peaking early in September, that is consistent with the timing of the species’ migration between breeding and wintering areas in Europe. The distribution of Nathusius’ pipistrelle is confined to a broad band from the British Isles in the west via continental Europe to the western fringe of Asia. The breeding areas are predominantly found in the northeastern and central parts of Europe. The wintering areas are mainly located at lower latitudes in southwestern and eastern Europe [40]. On the western board of the North Sea, the species hibernates on the British Isles [41]. The breeding areas are thought to be predominantly occupied by females. After the reproductive season, these areas are abandoned by the females and their offspring, to migrate to the wintering areas. Northeastern populations consist of long-distance migrants, fanning out to southwesterly directions during autumn migration [42,43]. To date, the longest known distance covered during migration is from Latvia to Spain, a distance of 2224 km [44]. In central Europe, the species is a short-distance migrant or even sedentary [45]. During migration, mating takes place along the migration routes, where males advertise to attract females [46]. After the mating season, part of the males migrate to the winter quarters at lower latitudes [45]. The timing of autumn migration shifts from a peak during the second half of August and the beginning of September along the Baltic sea coast in northeastern Europe [43] to September at lower latitudes, with, for instance, a peak in early September in the German Bight [27] and offshore locations in the North Sea [25,26,47]. In Great Britain, distinct peaks in occurrence in spring and autumn (September) can be explained by migration to and from continental Europe [41]. Therefore, we assume that the offshore occurrence off the Dutch west coast in autumn consists of migratory Nathusius’ pipistrelles attempting to cross the North Sea to wintering areas in Great Britain.

Although the study of bat migration is still in its infancy [48,49], theory on bird migration, which has been studied extensively, can be used as a framework for bats [50,51,52], since migration of both birds and bats consists of alternating periods of actual migratory flights and stopovers and depends on both intrinsic individual factors and multiple environmental factors (e.g., [52,53]). Before undertaking a long-distance migratory flight, an individual needs to adapt and transform from a sedentary individual, typically flying short distances, foraging and roosting at known places, to a migrant [54,55]. Once this ‘transformation’ has taken place, a migrating individual needs to have enough energy reserves to sustain migratory flights. Actual departure decisions to continue its migratory flight depend on weather conditions and on the availability of food.

As migrating bats do not store large fat reserves [56,57], their migration strategies differ in some aspects from those of the majority of migratory birds which do build up substantial pre-migration fat reserves. The most distinguishing difference with respect to birds is the use of torpor to minimize energy expenditure [56]. Furthermore, bats use a fly-and-forage strategy [58] in which the migratory flight is fueled directly from ingested insect prey, supplemented with deposited fat when prey is not sufficiently available [59]. Due to these strategies, migrating Nathusius’ pipistrelles probably do not need long stopovers to refuel. Prolonged stopovers probably occur only in adverse weather conditions, during periods with limited insect availability, or when faced with major ecological barriers [29,58]. Bats exhibit flexibility in using these migration strategies [57], balancing between minimizing energy expenditure and maximizing food intake to meet the changing and unexpected situations encountered during migration.

The timing of departures and stopovers is essential for the survival of migratory animals. Our analysis showed that wind speed, wind direction, and temperature are important explanatory variables for Nathusius’ pipistrelles’ autumn occurrence at sea. Nathusius’ pipistrelles occur offshore in correspondence of low to moderate wind speeds. The majority (67%) of our observations occurred with wind speeds < 5 m/s, an additional 31% occurred with wind speeds of 5–8 m/s, and the remaining 2% with wind speeds >8 m/s. These findings correspond with results from previous studies [26,27,29,60,61,62,63]. Wind speeds of 9–10 m/s, however, have been reported for Eastern Red Bats *Lasiurus borealis* off the United States’ east coast [64], showing that bats do fly in higher wind speeds over sea. Furthermore, we found that the highest probability of bat presence at sea occurred with wind from east-northeast, a direction that corresponds with the migration paths of virtually all ringing recoveries in Great Britain [65]. This indicates that migratory bats depart during optimal tailwind conditions and avoid crosswinds. A positive relation with tailwind conditions was also found off the eastern coast of the United States [64] and in the Belgian North Sea [63]. In the German Bight, however, bats were mainly observed during southerly winds in both autumn and spring, and the authors [27] suggested wind drift as a likely explanation for their offshore occurrence.

In addition to wind, temperature proved to be an important covariate. Average night temperatures > 15 °C were recorded during 89% of our observations. Land-based studies typically show a strong relationship between temperature and bat activity due to an increase in insect availability and activity triggered by higher temperatures (e.g., [66]). When this coincides with easterly winds, insects may drift offshore. Insects can also migrate in large numbers over sea, often at heights of several hundred meters above sea level [67,68,69]. Consequently, insect availability increases, enabling bats to fly-and-forage [58] during offshore migratory flights. Bats have been observed to interrupt their flight for foraging bouts around offshore wind turbines, where flying insects can accumulate [29].

Although previous studies identified cloud cover [27,60], lunar cycle [60], high atmospheric pressure [26,63], low atmospheric pressure [27,60], and rain [27] as important explanatory variables for bats occurring over sea, these covariates were excluded during the model selection, together with atmospheric pressure change and visibility. The spatial covariate XY coordinates was also excluded from our model, indicating that spatial differences in occurrence were not present in our study area. As bats are known to depart from specific coastal locations enabling them to minimize their flight distance over sea [29], spatial differences may be expected when studying the offshore occurrence at a larger scale.

Given the maximum flight speed of migrating Nathusius’ pipistrelle of 40–47 km/h [70] and the proximity of our monitoring locations to the mainland (<25 km from shore), we assume that we recorded bats that departed from the mainland the same night. Deteriorating weather conditions offshore or the arrival of daybreak may force bats to interrupt their flight and find a suitable structure at sea to roost, until weather conditions are suited to continue their journey, the next night or later. When this happens, recording the offshore presence of these bats depends on weather conditions during nights previous to the recording. Roosting bats have been occasionally observed by maintenance personnel at our nearshore monitoring locations in autumn (unpublished data). The lack of temporal autocorrelation in our data, however, supports our assumption that we recorded bats which started their offshore flight the same night.

Although continuous acoustic monitoring of Nathusius’ pipistrelle yields valuable information to elucidate their migration, our study has a few limitations. The first is inherent to acoustic monitoring in general [71,72], which lacks information to derive absolute numbers of individuals from bat detector recordings [73]. To obtain these absolute numbers, it is necessary to determine the detection probability of echolocation calls of bats, to quantify the size of the monitored area, and to obtain an estimate of a multiplier that translates the number of echolocation calls to an absolute number of individuals. This multiplier has to take into account that bats can reduce their echolocation output [74,75] or can even fly without echolocation for extended periods of time [76]. To account for this uncertainty, we used acoustic detections of bats expressed as presence/absence per night as a proxy for their occurrence, assuming the detection probability of an animal and the false detection rate are constant or at least random.

The height of our recorders (15 m above sea level) is the second limiting factor. Despite the fact that the detection range of the used equipment is not exactly known, it is restricted to a maximum limit of 30 m for pipistrelles [32]. This upper limit is based on individuals emitting their calls straight to the microphone, which will rarely occur in the field. In other words, echolocation calls can be detected up to ca. 45 m above sea level under optimal conditions only. Although information on flight altitudes is limited, the little we know points to low flight altitudes close to shore in the North Sea and Baltic Sea. In the North Sea, in a Belgian offshore wind park at 27 km from the coast, acoustic activity of Nathusius’ pipistrelles was significantly less recorded at nacelle height than at lower heights [63]. In the Baltic Sea, migrating bats were recorded predominantly <10 m, with a few individuals > 40 m. Altitudes > 100 m, however, were not covered [29]. These low flight altitudes are suggested to be restricted to coastal waters [27], whilst migration further offshore is suggested to be a high-altitude phenomenon, as shown during aerial surveys off the eastern coast of the United States [64]. At least 5 out of 11 diurnal eastern red bats were photographed between 16.9 km and 41.8 km offshore during tailwind conditions (9–10 m/s) at altitudes of more than 200 m above sea level, and one animal was recorded between 100 m and 200 m.

Furthermore, wind speed increases with height [77]. In our model, we used the wind speed measured at an altitude of 10 m above sea level. Assuming neutral atmospheric conditions, a wind speed of 5 m/s, for example, corresponds with a wind speed of 7 m/s at a height of 100 m [77]. The influence of wind on migrating bats, both the advantage of tailwinds and the disadvantage of headwinds, is likely to increase with height, as is shown for migratory birds and insects [52,78].

To conclude, our model results are only valid for low altitudes, since an unknown and variable proportion of migrating bats might pass undetected at high altitudes, using stronger tailwinds.

## 5. Conclusions and Recommendations

Monitoring acoustic bat activity in autumn at four monitoring locations 15–25 km from shore revealed a seasonal pattern that fits the timing of the Nathusius’ pipistrelles migration from the north-eastern breeding areas to the southwestern wintering areas. Migration explains the offshore occurrence of Nathusius’ pipistrelles off the Dutch coast and depends on low to moderate wind speeds, tailwind conditions to cross the North Sea to Great Britain, and higher temperatures. The latter possibly indicates a link with offshore insect availability, which may enable bats to fuel during their flight over sea. 

Although it should be noted that these findings are valid for altitudes below 45 m above sea level and that bats crossing the sea during migration may use higher altitudes and stronger tailwinds, our study provides valuable input to build a model to predict bat migration across the North Sea as an aid to prevent potential collisions of bats with offshore wind farms by taking mitigation measures.

To improve such a model, further bat detector research should focus on the offshore occurrence of bats on a broader spatial and temporal scale and on the flight distribution at different altitudes. In addition, telemetry studies can provide valuable information on actual migratory flight routes, as well on potential differences in migration strategies between sex and age classes.

## Figures and Tables

**Figure 2 animals-11-03442-f002:**
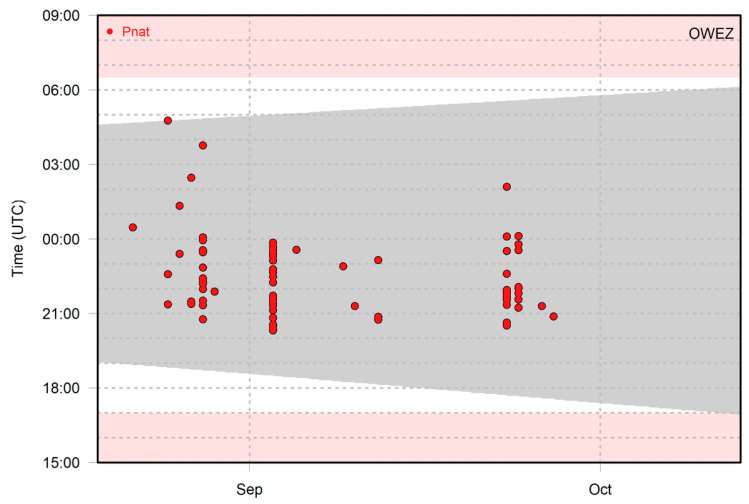
Occurrence of Nathusius’ pipistrelle in 1 min intervals at Offshore Wind Farm Egmond aan Zee in the period 20 August–13 October 2013. The grey area shows the time interval between sunset and sunrise. The white background shows the monitoring period, and the pink background shows the period with no monitoring/recorder switched off.

**Figure 3 animals-11-03442-f003:**
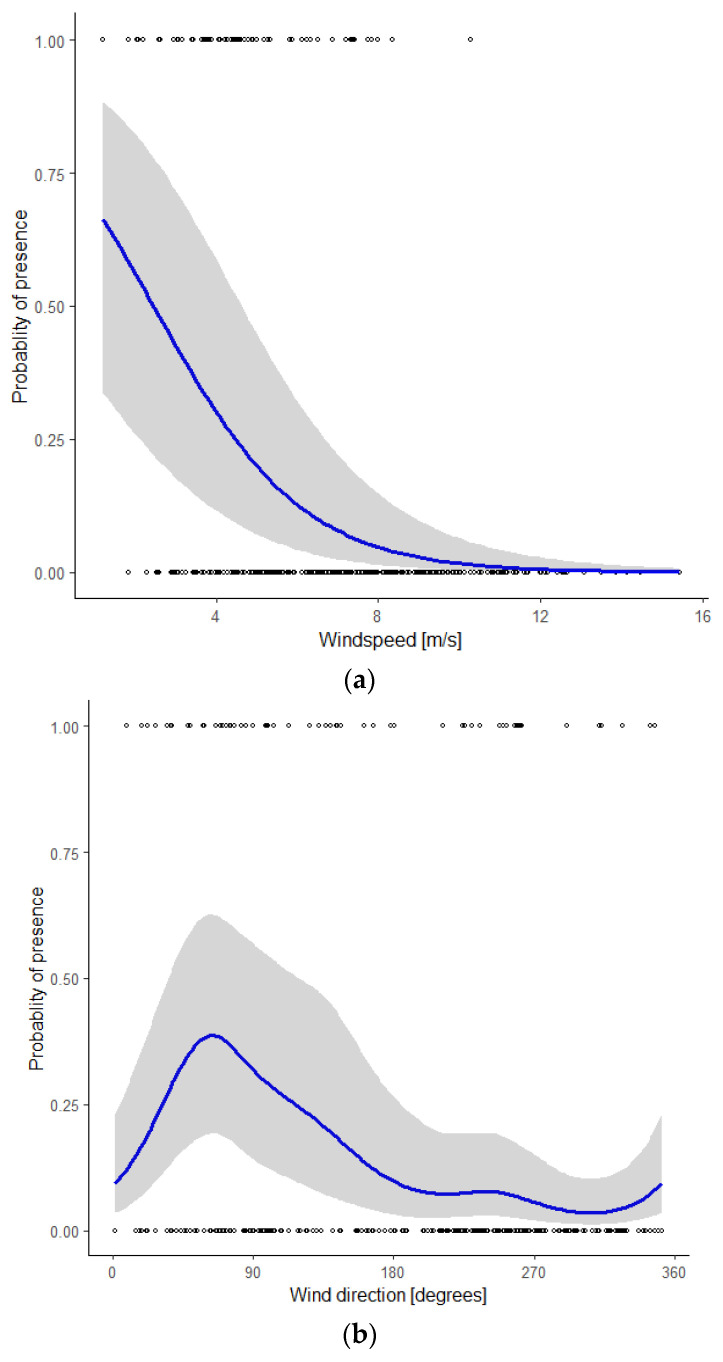
Probability of the presence of Nathusius’ pipistrelle as a function of the covariate (**a**) wind speed, given the mean values of the covariates wind direction, night in year, and temperature; (**b**) wind direction, given the mean values of the covariates wind speed, night in year, and temperature; (**c**) night in year (230 = mid-August, 285 = mid-October), given the mean values of the covariates wind speed, wind direction, and temperature; (**d**) temperature, given the mean values of the covariates wind speed, wind direction, and night in year.

**Table 1 animals-11-03442-t001:** Acoustic bat monitoring locations. Numbers correspond to location numbers in Figure 1. OWEZ mast = meteorological mast Offshore Wind Farm Egmond aan Zee, PAWP WT 22 and PAWP OHVS = wind turbine 22 and offshore high-voltage station Princess Amalia Wind Farm, and LUD OHVS = offshore high-voltage station Luchterduinen Wind Farm.

	Monitoring Location	Geographical Location	Distance to Shore [Km]	Heigh Above Sea Level [M]	Microphone Direction
1	OWEZ mast	52°61′ N, 4°39′ E	15	15	East
2	PAWP WT22	52°58′ N, 4°27′ E	23	15	East
3	PAWP OHVS	52°59′ N, 4°24′ E	25	15	East
4	LUD OHVS	52°40′ N, 4°17′ E	25	15	East

**Table 2 animals-11-03442-t002:** Monitoring periods and monitoring nights (N) in July–October per location per year. OWEZ mast = meteorological mast Offshore Wind Farm Egmond aan Zee, PAWP WT 22 and PAWP OHVS = wind turbine 22 and offshore high-voltage station Princess Amalia Wind Farm, and LUD OHVS = offshore high-voltage station Luchterduinen Wind Farm.

	OWEZ Mast	PAWP WT22	PAWP OHVS	LUD OHVS	Total
2012	29 August–20 October	4–23 September	-	-	
N	45	19	0	0	64
2013	1 July–15 October	5 August–2 October	-	-	
N	55	43	0	0	98
2014	1 July–14 October	-	1 July–15 October	-	
N	55	0	55	0	110
2015	-	-	1 July–20 October	1 July–9 October	
N	0	0	55	43	98
2016	-	-	1 July–17 October	1 July–24 October	
N	0	0	55	55	110
Total	155	62	165	98	480

**Table 3 animals-11-03442-t003:** Properties of parametric coefficients and smooth terms of the optimal model (AIC = 326.3676, df = 7, R-sq.(adj) = 0.345, glmer.ML = 275.61, Scale est. = 1, n = 478).

Optimal Model:				
Parametric Coefficients:	Estimate	Std. Error	*Z* Value	Pr (>|z|) ^1^
Intercept	−1.59188	1.69250	−0.941	0.3469
Windspeed	−0.54433	0.08546	−6.370	1.9 × 10 ^−10^ ***
Temperature	0.16386	0.09691	1.691	0.0908
Significance of smooth terms	Edf	Ref.df	Chi.sq	*p*-value ^1^
s (night in year)	3.533	3.533	18.39	0.000487 ***
s (wind direction)	4.130	8.000	36.05	<2 × 10^−16^ ***

**^1^** Signif. codes: 0 ‘***’ 0.001 ‘**’ 0.01 ‘*’ 0.05 ‘.’ 0.1 ‘ ’ 1.

## Data Availability

The data presented in this study are available on request from the corresponding author. The data are not publicly available yet, in due time they will be made publicly available in the datalab from the Dutch Offshore Wind Ecological Programme (WOZEP).

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
