# Peer review of "Offshore Occurrence of a Migratory Bat, Pipistrellus nathusii, Depends on Seasonality and Weather Conditions"

_animals, 2021, doi:10.3390/ani11123442_

Round 1

Reviewer 1 Report

Dear editor and authors,

This manuscript presents a study that relates nights with presence/absence of the migratory Pipistrellus nathusii bat to weather variables. Presence/absence nights were determined by acoustics measurements. The study is a welcome addition to a strongly understudied phenomenon (i.e., bat migration). The manuscript is generally well-written, and the methods are described clearly (hooray for that).

While I do think the manuscript is generally interesting enough for publication, I do have a couple of remarks that would probably improve the overall quality of the manuscript:

(1) Given the influence of weather on the sensitivity of the microphones, to which extent do you think this has influenced detection of bats. So, to which extent do you think that days of measured absence might actually be days of weather conditions not allowing to detect bats instead of bats not being present? (Note: I’m not an acoustics expert, so this is a genuine question.)

(2) Why did the authors choose to use stepwise backward selection? Were the parameter estimate of the full model too imprecise, i.e., with large uncertainty? Stepwise selection can be really problematic (even though unfortunately it’s still common practice).

Some refs of interest:
Whittingham, M. J., Stephens, P. A., Bradbury, R. B., & Freckleton, R. P. (2006). Why do we still use stepwise modelling in ecology and behaviour? Journal of Animal Ecology, 75(5), 1182–1189. https://doi.org/10.1111/j.1365-2656.2006.01141.x
Bolker, B. M., Brooks, M. E., Clark, C. J., Geange, S. W., Poulsen, J. R., Stevens, M. H. H., & White, J. S. (2009). Generalized linear mixed models: a practical guide for ecology and evolution. Trends in Ecology & Evolution, 24(3), 127–135. https://doi.org/10.1016/j.tree.2008.10.008

(3) Inference and prediction are two very different things. The former tries to explain the relation within the dataset, while the latter tries to estimate how a model will perform on previously unseen data. In particular, they need very different validation approaches to be able to make meaningful statements. To claim that you can now predict bat migration or occurrence (as is done a couple of times throughout the manuscript), you need to do some actual prediction exercises. Ideally, this is done through a repeated x-fold cross-validation in which you take into account temporal autocorrelation for the decision of how to create folds (see Roberts et al., 2017, for some more insights). Alternatively (albeit it perhaps less interesting), you can change the wording of the text to avoid saying that we can now predict when n.p. migration will occur across the North Sea.

To make things a lot more easier, you could use a random forest binary classifier for the prediction (instead of the gamm) and along with it you would also get an indication of variable importance (which would also address remark 2).

Roberts, D. R., Bahn, V., Ciuti, S., Boyce, M. S., Elith, J., Guillera-Arroita, G., Hauenstein, S., Lahoz-Monfort, J. J., Schröder, B., Thuiller, W., Warton, D. I., Wintle, B. A., Hartig, F., & Dormann, C. F. (2017). Cross-validation strategies for data with temporal, spatial, hierarchical, or phylogenetic structure. Ecography, 40(8), 913–929. https://doi.org/10.1111/ecog.02881.

(4) The authors will likely be very interested to also include the recent findings from Haest et al. (2021) on the drivers of migration phenology in Tadarida brasiliensis. They too found wind to have a strong effect on migration timing, albeit mainly in spring.

Haest, B., Stepanian, P. M., Wainwright, C. E., Liechti, F., & Bauer, S. (2021). Climatic drivers of (changes in) bat migration phenology at Bracken Cave (USA). Global Change Biology, 27(4), 768–780. https://doi.org/10.1111/gcb.15433

Detailed remarks

L15: i.c. -> i.e.

L41: should -> is likely to

L48: are being killed -> are killed

L39-44: Perhaps a bit much detail on electricity production capacity. Consider slimming down to 1 or 2 sentences.

L54-55: Sentence could use a reference to back up the statement.

L58-59: Sentence could use a reference to back up the statement.

L85-87:  Conclusive summary seems a bit out of place here at the end of the introduction. Consider deleting.

L94 onwards: Consider whether it is really worth creating acronyms for the locations instead of writing them in full across the manuscript. Non-generally used acronyms really make it more difficult for a reader to follow. The brain always needs to actively translate it. From experience with other MDPI journals, I also suspect word limit will not be a motivation to do so.

L124: “The monitoring periods in July-October per location are shown in Table 2.” -> Delete and put “ (Table 2).” At the end of the previous sentence.

L143-145: I would suggest to delete this also, and include the website as a reference which you can then cite in line 138 after the “29 May 2017”. People interested in using the same dataset will for sure find their way to it like this.

L153 – formula 1: Formula is rather hard to read. Seems like R code instead of a formula. You should probably try to write it more like a mathematical formula.

L292: Yes, very much, but see Haest et al. (2021) and refs therein.

L328: Is [52] the correct reference here?

L343-344: Well, most of it will be explained by the random site effect also, right? So, perhaps not so surprising the fixed effects for lat and lon get dropped. Btw: I don’t see the random effect of site in Table 4. This needs to be included also.

L358-368: This should be included in the methods under section 2.2 or 2.3 instead of here.

L381: Reference need to be adjusted to journal’s format.

L387: “Wind speed, … “: Such a statement needs a ref… The direction could actually also be in the other direction at different altitudes…

L405: predict -> to predict (but see my main comment on prediction)

Reviewer 2 Report

This manuscript on the occurrence of bats in offshore windparks in relation to the prevailing weather conditions is interesting to read and offers new and important insights for a better protection of this animals in the future. However, there are a number of issues which need clarification prior to publication of this results as stated below:

The introduction focuses strongly on offshore wind. However, this is not reflected in the title of the paper, the statistical analysis and the discussion, where the focus lies on offshore occurrence of bats throughout the migratory season and with regard to weather. I believe some of the information on the actual animals needs to be moved from the discussion to the introduction. On the other hand, the discussion totally lacks a relation to offshore wind farms, like e.g. if bats occur offshore predominantly under very low wind speeds, are the wind farms even operating? What does it tell you that the bats are so frequently recorded at receivers mounted below 30m in your study? Do they probably stay below the actual operation heights of the rotors? If you want to focus on offshore wind as much as stated in the introduction, I would expect such questions to be addressed. Still I would rather see the focus to be shifted towards your actual investigations of bat behaviour with regard to weather.

Line 11: Should read „conditions when bats can be expected“

Line 15: the term tailwind should be avoided throughout the manuscript as the direction the bats are flying to or aiming at can only be guessed.

Line 19: should read “in offshore windfarms”

Line 44-49 please state clearly at all times which of the data stem from offshore/ onshore investigations.

Line 48: Perhaps rephrase to “are likely being killed” as these are not actual counts of dead bats but extrapolations?

Line 53 not being an expert on wind farm operation, I would like to get a little more info on what curtailment-procedures and cut-in-speeds are and how these help the bats.

Line 71 please rephrase to “ conditions under which bats”

Line 83 Only in the discussion an explanation is given for how bats could be present at sea without having departed from the coast within the same night. I am wondering how common it actually is that bats sleep at offshore wind platforms? I would suggest to leave this confusing sentence out at this point, but if you consider it necessary in the introduction, please move the according information from the discussion.

Line 85 ff This sentence should be moved to the discussion

Fig 1: Please explain the symbols (stars, red dot..) used in the figure. A scale in the inlayed map is missing.

Line 135 If the data you use are simple presence/absence, why was it necessary to identify taxonomy? Also nowhere in the results information on what species you identified is given. Later on you are only discussing Nathusius pipistrelles like the dataset was restricted to those animals? Are they the only species you identified? Are there others? Could a species per night kind of info be added to the models?

Line 137 ff some important information on the weather measurements are not given in this paragraph but in Line 167 ff. Please join this information here

Line 147 What are non-satisfactory data? And if data from a single station are not good to use, why did you use them in those cases were only at one station data are available? How often did that happen? Could you include the location of the weather stations in Fig. 1?

Line 166 ff What are these formulas? What do the included terms mean? Only formula (1) is explained in the text and only (5) seems to be the model structure you are using?

Line 167 I was surprised that the authors chose to include the x and y coordinates of the recorder sites in their GAMM model along with the different study sites included as random effect. It’s like controlling for effects of the different sites while at the same time testing for them? With just the information on the spots location from the map I would suggest to remove the coordinates from your model or give a reason why you chose to include them. However, in terms of interpretation, this is a minor problem, as the coordinates are not part of the final model anyways.

Did you use knots to define the circularity of the wind data? In figure 3 it looks like there is a small gap between the actual data/ model prediction and 360°.

Line 267 ff Information on the migratory ways of animals in the study area might be given in the intro rather than the discussion. Other information like maximum distance covered and mating seem not to be related to the present paper at all? In my opinion the discussion might as well start in line 281. The focus on Nathusius pipistrelle is confusing as the actual data you analyse are from different species and this is not mentioned anywhere in the discussion?

Line 286 please add the information that Helgoland is a remote island. Why is it separated from “offshore locations in the North Sea”?

Line 301 largely true but not solely as the internal state of an animal like for example it’s state of health may also play a role.

Line 345 What do you mean by specific locations? Specific with regard to..?

Line 349 as previously said, if sleeping offshore is a common behaviour in bats, this should be stated earlier!

Line 375 Could you give an estimate for how high above bats could still be seen flying with this method? Otherwise it is hard to judge the given information.

Line 381 wrong citation format

Line 399 makes me wonder: would it be possible to give a map on the breeding and wintering locations of bats passing through the study area? Perhaps added to a larger version of the now inlayed map in figure 1?

Line 405 should read “input to predict”

Round 2

Reviewer 1 Report

Dear editor,

I have no further major remarks.

The authors should carefully check the citation numbers though. I identified at least 1 more citation that seems to the wrong number (l. 350).

In Table 1, geografical -> geographical

All the best & congratulations.

Author Response

Thank you again for your time and valuable comments that improved the paper.

We corrected the typo and checked the references. Furthermore, we have thoroughly edited our English  and made the structure of the discussion clearer.

Sander, Bob & Steve

Reviewer 2 Report

Dear Editor and Authors,

Thank you for considering my comments and suggestions on the first version of the manuscript. In my first review I commented on a misbalance of a very detailed focus on offshore wind farms in the introduction and not mentioning this topic at all in die discussion part of the manuscript. This issue unfortunately has not been addressed and remains present in the revised version of the manuscript. The authors commented that their study in fact raised new questions which they recommended as follow up studies. I would have hoped that the authors at least include these open questions and point out the need to address those in future studies in the discussion. Such information could also lead to the conclusion of building a model to predict bat migration in the future, which now pretty much stands alone. Please find a few more minor comments to the revised manuscript version below.

Line 16: Please add the actual wind directions as done in the abstract

Line 71: In the North-Sea is strange. Perhaps you could use “flying over” or a related phrase.

Fig. 1: A scale in the inlayed map is still missing

Line 146 f Now that you clarified to only use recordings of Nathusius pipistrelles I expected to also see this sentence changed by either briefly adding info on the other taxa to the results section or only referring to the identification of Nathusius pipistrelle here.

Line 187 ff: In my first review I stated that I did not understand why both the monitoring location and the XY coordinates of the bat detectors were used in the same model. The authors clarified in their reply that the monitoring location contains various more information. Sorry, but I do not understand this statement. Does this mean the location is not included as one but multiple variables in the model? From what is written in the paper I would suggest it to be a factor including the names of the single sites. As this seems not to be the case, please add this information in full transparency to the manuscript

Line 201: I am aware you are using a circular smoother. Still, using only the smoother to my knowledge connects the smallest to the largest data point. So if you do not have values of exactly 0 and 360 degrees in the dataset, which is what it looks like in the figure, the smoother will work inaccurately. To correct for this you can specify knots to add the information to the model that your data range between 0 to 360 degrees. Still, as you do obviously have values at least rather close to both values, this is probably more a technical detail and will not substantially change the outcome of this specific analysis.

Lines 292 ff I am still not convinced that all the given information in this paragraph is needed. You give information on behavioural differences of male and female bats and shed light on mating behaviour. Still, how is this information linked to your analysis? There is no information given on whether you recorded mating calls or how offshore-movements might be related to sex specific behavioural differences. Please clarify this linkage or remove this information from the discussion.

Line 371 please add that the bats seem to prefer such spots as now it sounds like they only ever choose such spots to fly out to sea

Line 374 ff You might want to resort this paragraph now that the introduction part on bats staying over day at offshore platforms has been removed. Starting with the statement that bats do sleep on offshore structures and then stating why you think this can be at best neglected in your analysis would at least be easier to follow for me as a reader.

Author Response

Thank you very much for taking your time to review our revised manuscript. We have adressed your comments as good as possible. Please see the attached file for a detailed response to your comments.

Sander, Bob & Steve
